# WATT FOR WHAT: RETHINKING DEEP LEARNING'S ENERGY-PERFORMANCE RELATIONSHIP

## ABSTRACT

Deep learning models have revolutionized various fields, from image recognition to natural language processing, by achieving unprecedented levels of accuracy. However, their increasing energy consumption has raised concerns about their environmental impact, disadvantaging smaller entities in research and exacerbating global energy consumption. In this paper, we explore the trade-off between model accuracy and electricity consumption, proposing a metric that penalizes large consumption of electricity. We conduct a comprehensive study on the electricity consumption of various deep learning models across different GPUs, presenting a detailed analysis of their accuracy-efficiency trade-offs. By evaluating accuracy per unit of electricity consumed, we demonstrate how smaller, more energy-efficient models can significantly expedite research while mitigating environmental concerns. Our results highlight the potential for a more sustainable approach to deep learning, emphasizing the importance of optimizing models for efficiency. This research also contributes to a more equitable research landscape, where smaller entities can compete effectively with larger counterparts. This advocates for the adoption of efficient deep learning practices to reduce electricity consumption, safeguarding the environment for future generations whilst also helping ensure a fairer competitive landscape.

## 1 INTRODUCTION

Deep learning has emerged as a powerful technology, achieving remarkable breakthroughs across various domains. From image recognition and natural language processing to autonomous driving and healthcare diagnostics, deep learning models have redefined the boundaries of what machines can accomplish. The stunning advances in accuracy have transformed industries, offering new solutions to long-standing problems.

However, these strides in deep learning come at a significant cost, both in terms of energy consumption Desislavov et al. (2021) and environmental impact Selvan et al. (2022). As models grow larger and more complex, they demand increasingly substantial computational resources. This insatiable appetite for electricity not only drives up operational costs but also has alarming implications for the environment. The environmental footprint of training large-scale deep learning models is substantial Anthony et al. (2020); Strubell et al. (2019), contributing to growing concerns about climate change and resource depletion.

In a landscape where cutting-edge deep learning research often necessitates access to colossal computational infrastructure, small companies and academic institutions find themselves at a disadvantage. Competing with tech giants and well-funded organizations on the basis of computational resources alone is an unattainable goal for many. This stark inequality not only hampers innovation but also perpetuates a power imbalance in the field, potentially biasing progress toward the economic interests of these dominant companies.

The current environmental situation is dire, with climate change accelerating, species going extinct at alarming rates, and pollution threatening human health around the globe. Meanwhile, the growth of deep learning and AI is exacerbating these environmental crises. Training complex AI models requires vast amounts of data and computing power, consuming massive amounts of electricity. For example, training a single large language model can emit as much carbon as a car over its lifetime Bannour et al. (2021). As these models get even bigger, their energy consumption and carbon

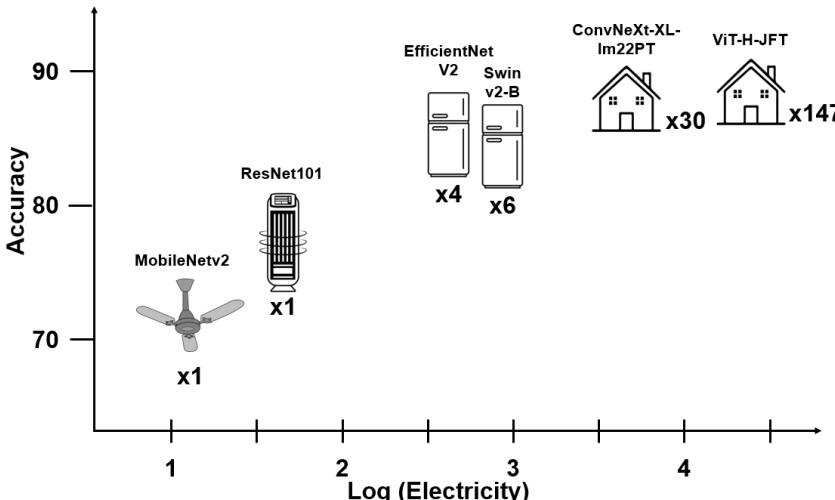

Figure 1: Bridging the Energy Divide: Deep Learning Models vs. Everyday Power Hogs. For easy comparison, we list the amount of electricity consumed per month by an appliance or by the average household in the UK.

emissions skyrocket. Take for example, a Boeing 747 flying from Heathrow to Edinburgh, a distance of 530 kilometers, emits approximately 400 tonnes of CO2 emissions while training the GPT model in one of the most carbon-intensive areas results in emissions of approximately 200 tonnes of CO2 Khowaja et al. (2023). In terms of electricity consumption, the OPT-175B Meta model consumes a staggering 356,000 kWh of electricity during training and the GPT-3-175B model developed by OpenAI consumes an astonishing 1,287,000 kWh of electricity during its training process Khowaja et al. (2023). For reference, a hypermarket in the UK (average area of 6262 square metres) uses 2,523,586 kWh electricity in a year Kolokotroni et al. (2019).

This is unsustainable at a time when we urgently need to cut emissions and transition to clean energy. While AI and deep learning offer many benefits, their development must become far more energy-efficient. Researchers must prioritize creating more efficient methods and hardware so that AI's growth does not continue worsening environmental degradation. With climate change accelerating, we need to rein in AI's energy and data appetites before it's too late.

To address this issue, this paper takes a critical step towards achieving a more balanced and sustainable approach to deep learning research. We investigate the trade-off between model accuracy and electricity consumption, introducing a metric that levels the playing field for all participants in the deep learning arena. We take image classification as the main task to study this trade-off, considering models from VGG16 Simonyan & Zisserman (2014b) to the latest transformer models Dosovitskiy et al. (2020); Touvron et al. (2021); Liu et al. (2021), including self-supervised He et al. (2022); Radford et al. (2021) learning models. Additionally, we extend our analysis to image segmentation and video action recognition, taking into account not only the cost of training the models themselves but also the pre-training costs.

By evaluating accuracy per unit of electricity consumed, we also empower smaller entities, including universities and startups, to compete effectively with industry leaders. Furthermore, we advocate for the adoption of energy-efficient model architectures, which not only reduce electricity consumption but also expedite research, enhancing overall efficiency. In Figure 1, we look at some deep learning models and their cost in comparison to a real-life estimate using similar electricity. We compare a few deep learning models with day-to-day appliances and for the larger models estimate it with respect to the electricity consumption of an entire household BritishGas (2023).

This paper presents an in-depth analysis of deep learning model efficiency and its implications for research, industry, and the environment. It emphasizes the need for a fundamental shift in the way we measure and optimize deep learning models, highlighting the importance of sustainability in the era of AI.

## 2 RELATED WORK

### 2.1 WORKS ABOUT EFFICIENCY

Improving the efficiency of neural networks has become an important area of research in deep learning. As neural networks have grown larger in size and complexity, their computational costs for training and inference have also increased. This has led to novel techniques that optimize neural network efficiency without significantly sacrificing accuracy.

One approach is pruning Blalock et al. (2020), which removes redundant or non-critical connections in a trained neural network. Studies have shown large parts of neural networks can be pruned with minimal impact on performance. This leads to smaller, faster models. Another technique is quantization Weng (2021), which converts 32-bit floating point weights and activations to lower precision representations like 8-bit integers. While quantization can lead to some loss in accuracy, re-training the quantized model can help recover the lost accuracy.

Another family of methods is efficient neural architectures, such as depth-wise separable convolutions used in Xception Chollet (2017) and MobileNets Howard et al. (2017), or using squeeze-and-excitation Hu et al. (2018) blocks which require fewer computations. Knowledge distillation Hinton et al. (2015) is another efficiency technique, where a small and fast student model is trained to mimic a large teacher model, allowing the student to achieve comparable accuracy.

Architectures for video understanding face high computational costs due to complex operations handling spatiotemporal information Tran et al. (2015); Carreira & Zisserman (2017b). Alternative approaches include 2D CNN-based models Simonyan & Zisserman (2014a), temporal shift modules Lin et al. (2019), and decomposing spatiotemporal information into multiple subspaces Feichtenhofer et al. (2019); Pan et al. (2021), focusing on more efficient architectures without considering input video characteristics. Certain action recognition methods attain efficiency by selectively choosing a subset of frames from an input video for prediction, by utilizing either a lightweight network Korbar et al. (2019); Gowda et al. (2021) or multiple reinforcement learning agents Wu et al. (2019; 2020) to determine the frames to be passed into the full backbone models.

Defining efficiency in neural networks is nuanced, as there are multiple cost indicators that can be considered, such as FLOPs, inference time, training time, and memory usage Dehghani et al. (2021). Improvements in one dimension, such as FLOPs reduction, do not necessarily translate to better efficiency in other dimensions like training time or memory usage. As such, it is important to consider the specific goals and hardware constraints when evaluating and comparing the efficiency of neural network models.

### 2.2 ENVIRONMENTAL IMPACT

Research on the environmental impact of deep learning models is an exciting emerging field. Ligozat et al. (2022) reviews existing tools for evaluating the environmental impacts of AI and presents a framework for life cycle assessment (LCA) to comprehensively evaluate the direct environmental impacts of an AI service. It highlights the need for energy-efficient algorithms and hardware to reduce AI's environmental footprint. However, this work only talks from a theoretical perspective without diving deep into the actual numbers involved.

Large deep learning models can have high computational costs during training, leading to significant carbon emissions and energy usage Strubell et al. (2019). For example, Strubell et al. (2019) estimated that training a large transformer-based language model can emit as much as 626,000 pounds of carbon dioxide, equivalent to nearly 5 times the lifetime emissions of an average American car.

Several studies have investigated methods to reduce the carbon footprint of deep learning. Schwartz et al. (2020) examined techniques like early stopping and neural architecture search to optimize and reduce training compute. Bender et al. (2021) proposed that model card documentation should include details on compute infrastructure and carbon emissions.

There is also research quantifying the electricity usage and carbon emissions of various models. Lacoste et al. (2019) measured the energy consumption of common AI tasks on hardware like CPUs and GPUs. Patterson et al. (2021) estimated training the GPT-3 model emitted over 552 tonnes of

carbon dioxide. Both studies emphasize the importance of hardware efficiency and carbon accounting in deep learning.

Optimizing model architecture, training procedures, and hardware efficiency are important ways researchers are working to limit the environmental impacts of deep learning. More transparency around carbon emissions and energy use will also help advance sustainability efforts.

### 2.3 COMBATING THE IMPACT

Yarally et al. (2023) highlighted the importance of balancing accuracy and energy consumption in deep learning. By analyzing layer-wise energy use, they showed model complexity reduction can significantly lower training energy. This work advocates energy-efficient model design for sustainability. Getzner et al. (2023) proposed an energy estimation pipeline to predict model energy consumption without execution. However, GPU measurement was not covered and CPU-based estimation has limitations.

In contrast to prior studies, our research offers a comprehensive analysis spanning from foundational deep learning models like VGG16 to the latest iterations of the vision transformer. Our evaluation extends beyond image classification, encompassing tasks like action recognition and semantic segmentation, thus providing a more extensive assessment. Recognizing the resource disparities between smaller university labs and companies in comparison to larger enterprises, we introduce a novel metric that factors in a deep learning model's power consumption alongside traditional performance metrics like accuracy or mIoU. This metric aims to level the playing field, acknowledging that individuals from smaller organizations may face hardware limitations that preclude them from training exceptionally large models.

### 2.4 TRACKING POWER CONSUMPTION

A number of easily available open-source tools help with the goal of tracking emissions and electricity consumption. CodeCarbon Henderson et al. (2021) is an open-source Python package that can estimate the $CO_2$ emissions and electricity usage of running code, including ML training and inference. It integrates with popular libraries like PyTorch and TensorFlow and can be used as a decorator to seamlessly quantify emissions. Similarly, CarbonTracker Anthony et al. (2020) instruments ML code to monitor energy consumption throughout experiments, giving insights into the most carbon-intensive parts of model development. Other popular tools include TraCarbon Valeye (2023), Eco2AI Budennyy et al. (2022) among others.

We use CodeCarbon for all our experiments and show linear dependency between data and number of epochs/iterations in our experiments (please see Appendix). Based on this, authors would only need to train on 1% of data and 1 epoch to scale up the estimated power consumption, making it an easy and non-tedious job to do.

## 3 PROPOSED METRIC

Achieving high accuracy may require substantial computational resources, creating a trade-off between performance and environmental sustainability. In this context, there is a clear need for a comprehensive metric that not only evaluates the error rate but also accounts for power consumption. Such a metric would encourage responsible resource usage, promote energy-efficient computing practices, and foster fair comparisons between different systems or models.

We propose 'GreenQuotientIndex (GQI)' to help us with this. GQI is defined in Eq. 1. Where '$accuracy$' refers to the accuracy rate (in range 0 to 1), '$electricity$' is the total electricity consumed in KwH and '$\alpha$' and '$\beta$' are constants that help in scaling up the GQI for better understanding of the values. We set '$\alpha$' and '$\beta$' to 5 based on our experiments and discuss this in Section A.5 in the Appendix. GQI effectively computes the cost of power per accuracy percentage point. Although the most straightforward way to compute this would be a simple ratio of accuracy divided by power, this has two issues. The first is that power consumption varies widely across models, making comparison difficult. Thus, we take the logarithm of the power consumption to make values more easily comparable. The second issue is that not all accuracy points are created equal, and compute and

accuracy follow some sort of Pareto principle [1]. In general, it is easier to improve a method from 30 to 40% than it is to improve from 80 to 90%. In other words, it is much easier to correctly classify some parts of the dataset than others. Therefore, the metric needs to take into account this difference and reward those approaches that achieve very high accuracy. We do this by taking the power of the accuracy.

$$GQI = \beta \times \frac{accuracy^{\alpha}}{log_{10}(electricity)} \tag{1}$$

Overall, GQI effectively penalizes high electricity consumption while simultaneously reinforcing the importance of accuracy improvements, thereby encouraging innovation in more sustainable model development without undermining the valuable contributions of cutting-edge technologies like vision transformers.

## 3.1 PROPERTIES OF THE METRIC

- **Range**: Range for denominator is from [0, inf], range for numerator is from [0, inf], therefore the range for GQI is [0, inf].

- **Scaling Down**: The metric scales down both accuracy rate (as it is a power function of a number less than 1) and electricity consumption through logarithmic transformations, ensuring that the values are on a consistent scale for meaningful comparison.

- **Penalizes High Power Consumption**: It penalizes models with high power consumption, promoting energy-efficient and environmentally responsible computing practices.

- **Promotes Accuracy**: While penalizing power consumption, the metric still encourages improvements in accuracy by considering the trade-off between error rate and energy use.

- **Non-Negative**: The metric is always non-negative, which is an important property for any evaluation metric, as negative values wouldn't make sense in this context.

- **Comparability**: It enables fair comparisons between different computational systems or models, facilitating informed choices based on efficiency and sustainability.

- **Sensitivity**: The metric is sensitive to variations in both error rate and power consumption, making it suitable for detecting small changes in system performance and power usage.

- **Resource-Agnostic**: The metric is versatile and can be applied to a wide range of computational systems or models, regardless of the specific technology or domain.

## 4 ENERGY CONSUMPTION ACROSS TASKS

We address a range of tasks encompassing image classification, semantic segmentation, and action recognition in videos. Given that the majority of these tasks employ an ImageNet-pretrained backbone, we designate ImageNet as the dataset for image classification. Additionally, we extend our analysis to encompass semantic segmentation and action recognition tasks, evaluating the overall costs by factoring in both pretraining costs and the expenses associated with training the models.

## 5 EXPERIMENTAL ANALYSIS

### 5.1 IMAGE CLASSIFICATION

In our study, we employ an extensive array of models and methodologies to comprehensively evaluate performance across tasks. These include well-known architectural models like MobileNetv2 Sandler et al. (2018), MobileNetv3 Howard et al. (2019), ResNet He et al. (2016) series, VGG16 Simonyan & Zisserman (2014b), Inception v3 Szegedy et al. (2016), RegNetY Radosavovic et al. (2020) and DenseNet Huang et al. (2017), as well as more recent advancements such as EfficientNet v1 Tan & Le (2019), v2 Tan & Le (2021) and ViT (Vision Transformer) Dosovitskiy et al. (2020) models.

---

[1] https://en.wikipedia.org/wiki/Pareto_principle

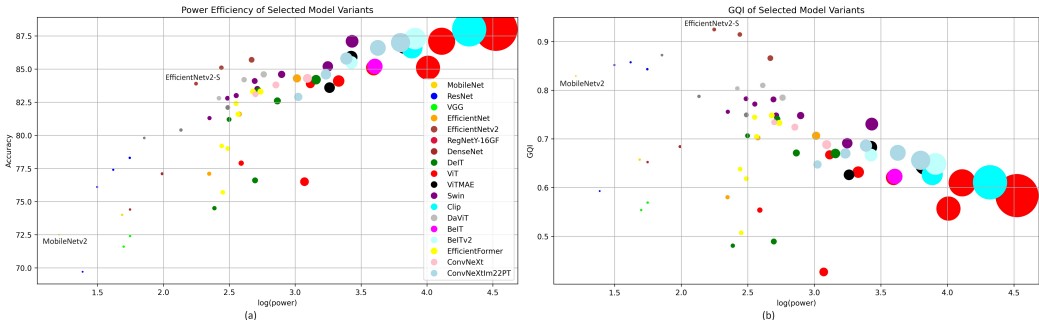

Figure 2: Here, we see how the proposed GQI helps achieve a better trade-off between accuracy and electricity used. In (a) we see models like Mobilenet, ResNet, EfficientNet etc much lower in the graph due to their relatively lower accuracy in comparison to models like ViT-H. Using the GQI, in (b) we see models like ViT-H and CLIP being penalized for the huge amounts of electricity being consumed and models like EfficientNet and MobileNet scaling much higher in the graph. The marker points for each model is scaled up to reflect the amount of electricity consumed (Zoom in for better clarity).

We incorporate novel approaches like Swin Transformers Liu et al. (2021), DeIT (Data-efficient Image Transformer) Touvron et al. (2021), DaViT (Dual Attention Vision Transformers) Ding et al. (2022), and BEiT (BERT Pre-training of Image Transformers) Bao et al. (2021), EfficientFormer Li et al. (2022) and ConvNeXt Liu et al. (2022), along with their adaptations to pretraining (ImageNet21k Ridnik et al. (2021) or JFT Sun et al. (2017)). Notably, we also investigate CLIP (Contrastive Language–Image Pre-training) Radford et al. (2021) and MAE (Masked Autoencoders) He et al. (2022) techniques to encompass a broad spectrum of cutting-edge methods for thorough evaluation.

We use 1 NVIDIA RTX A6000 48 GB GPU for all experiments in this scenario. We use 10 % of the ImageNet Deng et al. (2009) dataset randomly sampled and use a fixed batch size of 32 for all experiments. Image size is fixed to the size used in the original implementations. All other hyperparameters are also fixed based on the source paper. We use the Pytorch Image Models (timm) [2] for all implementations.

In Figure 2 (a), we have plotted model accuracies on the x-axis against the logarithm of each model's electricity consumption on the y-axis. Lower values on the y-axis indicate greater efficiency. The size of each marker in the plot is determined by the model's electricity consumption, making models with higher electricity usage appear significantly larger in scale. This visualization clearly illustrates a pattern where higher accuracy is associated with models that consume more electricity. However, as discussed earlier, this trend is not practical. On the other hand, Figure 2 (b) introduces a novel metric known as the proposed GQI, which dramatically shifts this pattern. The GQI penalizes models that require a substantial amount of electricity to achieve only marginal gains in accuracy. Consequently, in this graph, we observe that models such as MobileNet and EfficientNet perform much better, highlighting the crucial significance of efficient training and model architecture. Given the current environmental concerns and the challenges of competing with resource-rich companies, we believe that adopting such a sustainability-focused metric is essential for responsible research advancement. For a more detailed breakdown of the data presented in Figure 2, including information on energy consumption, accuracy, and the proposed GQI, please refer to the Appendix, Table A.1.

## 5.2 SEMANTIC SEGMENTATION

For semantic segmentation, we conduct experiments on a number of representative approaches. Specifically, these include two classical methods PSPNet Zhao et al. (2017) (backbone ResNet101) and DeepLabv3 Chen et al. (2017) (backbone ResNet101), the real-time segmentation model BiseNet Yu et al. (2018) (backbone ResNet18), ViT-based architectures Segmenter Strudel et al. (2021) (backbone ViT-L) and SETR Zheng et al. (2021) (backbone ViT-L), a universal segmenta-

---
[2]https://huggingface.co/timm

| Dataset | Models | Train | | | | Test | | |
|---------|--------|-------|-----------|-------------|------|------|-------------|
| | | GFLOPs | Parameters | Electricity | mIoU | GQI | Electricity |
| Cityscapes | PSPNet | 256G | 65.6M | 67.9001 | 80.2 | 0.906 | 0.019259 |
| | DeepLabv3 | 348G | 84.7M | 79.5025 | 81.3 | 0.935 | 0.031512 |
| | BiSeNet | 14.8G | 13.3M | 27.0179 | 77.7 | 0.989 | 0.004970 |
| | Segmenter | 400G | 334M | 1243.6126 | 81.3 | 0.574 | 0.074713 |
| | SETR | 417G | 310M | 1243.4491 | 81.6 | 0.585 | 0.073880 |
| | Mask2Former | 90G | 63M | 48.5331 | 82.2 | 1.113 | 0.005852 |
| ADE20K | BEiT | 605G | 162M | 1562.5031 | 45.6 | 0.031 | 0.053533 |
| | MAE | 605G | 162M | 1902.9770 | 48.1 | 0.039 | 0.035615 |

Table 1: A comparison of electricity consumed, accuracy and the proposed metric over multiple models. When we look at models using 2D CNN backbones, we see that they use much less electricity compared to ViT backbones. Surprisingly, they can still compete with and sometimes even outperform ViT models. Our GQI analysis confirms this by ranking these methods much higher in terms of efficiency and performance.

tion model Mask2Former Cheng et al. (2022) (backbone ResNet50), and two self-supervised learning models BEiT Bao et al. (2021) and MAE He et al. (2022) (both using ViT-B backbones).

We run all experiments for 2k iterations on 4 NVIDIA V100 32GB GPUs with batch size 8. All implementations are based on the opensource toolbox MMsegmentation [3] and two widely used segmentation datasets, Cityscapes Cordts et al. (2016) and ADE20K Zhou et al. (2017). FLOPS are estimated using an input size of $3 \times 512 \times 512$.

In Table 5.2, we present a comparative analysis of power efficiency among selected model variants for segmentation. Examining models with 2D CNN backbones, we find that they consume far less electricity than ViT backbones, yet they can still rival or even surpass ViT models in performance. Our GQI analysis underscores this by giving these methods a significantly higher ranking.

## 5.3 ACTION RECOGNITION

In this section, we perform experiments to evaluate the performance in the video action recognition task. Our evaluation encompasses various models, including the TimeSformer Bertasius et al. (2021) model based on the ViT architecture, classical CNN-based models like MoViNet Kondratyuk et al. (2021), I3D Carreira & Zisserman (2017a), TSM Lin et al. (2019), and TRN Zhou et al. (2018), as well as the UniFormerV2 Li et al. (2023) model, which combines the advantages of ViTs and CNNs.

All the experiments in this task were carried out using 2 NVIDIA RTX 3090 24GB GPUs. We configured the batch size as 8, keeping all other hyperparameters and image sizes consistent with those outlined in the source paper. We utilized the official implementations in this task. The experiments were evaluated on two widely used datasets: Kinetics-400 Kay et al. (2017) and Something-Something V2 Goyal et al. (2017).

In Table 5.3, we present a comparative analysis of power efficiency among selected model variants in the context of action recognition. We compare the electricity consumed with the GQI. By using the GQI, we manage to strike a more favorable balance between accuracy and electricity consumption. For instance, in the context of the SSv2 dataset, which is characterized by significant temporal classes, TSM emerges as the top-performing model, primarily because it consumes minimal electricity despite its lower accuracy compared to Uni v2 (CLIP). However, the scenario changes when we consider the Kinetics dataset, where there is a more substantial gap in accuracy, leading to TSM securing a lower ranking.

## 5.4 DATA VS POWER CONSUMPTION

One evident aspect of pre-training on extensive datasets is the significant electricity consumption it entails. Nevertheless, in light of the concerns regarding the limited availability of public data (such as the JFT dataset or the dataset used by CLIP), the lack of access to the necessary hardware for

---

[3]https://github.com/open-mmlab/mmsegmentation

| Dataset | | Timesformer | Uni V2 (IN21K) | Uni V2 (CLIP) | MoviNet | I3D | TSM | TRN |
|---|---|---|---|---|---|---|---|---|
| | GFlops | 590G | 3600G | 3600G | 2.71G | 65G | 65G | 42.94G |
| | Params | 121.4M | 163.0M | 163.0M | 3.1M | 12.1M | 24.3M | 26.64M |
| SSv2 | Train Ele | 1328.0169 | 1421.4425 | 7614.0746 | 303.3792 | 118.4572 | 107.3610 | 100.5043 |
| | Acc | 59.5 | 67.5 | 69.5 | 61.3 | 49.6 | 63.4 | 47.65 |
| | Test Ele | 0.2245 | 0.2899 | 0.2899 | 1.9653 | 1.9604 | 0.5178 | 0.2569 |
| | GQI | 0.119 | 0.222 | 0.209 | 0.174 | 0.072 | 0.252 | 0.061 |
| K400 | Train Ele | 1337.1037 | 1466.8697 | 7632.2455 | 421.3966 | 136.6454 | 124.6118 | — |
| | Acc | 78.0 | 83.4 | 84.4 | 65.8 | 73.8 | 74.1 | — |
| | Test Ele | 0.1790 | 0.2312 | 0.2312 | 1.5673 | 1.5634 | 0.4128 | — |
| | GQI | 0.462 | 0.637 | 0.552 | 0.235 | 0.513 | 0.533 | — |

Table 2: A comparison of electricity consumption, accuracy, and the proposed metric across multiple models in the action recognition task on the Something-something v2 dataset and the Kinetics-400 dataset. Timesformer corresponds to the divided space-time attention and Uni V2 (IM21k) uses a ViT-B pre-trained on IM21k and similary Uni V2 (CLIP) corresponds to the pre-trained CLIP model. 'Train Ele' and 'Test Ele' corresponds to the electricity consumed at train and test time respectively. Using GQI we achieve a better trade-off between accuracy and electricity consumed.

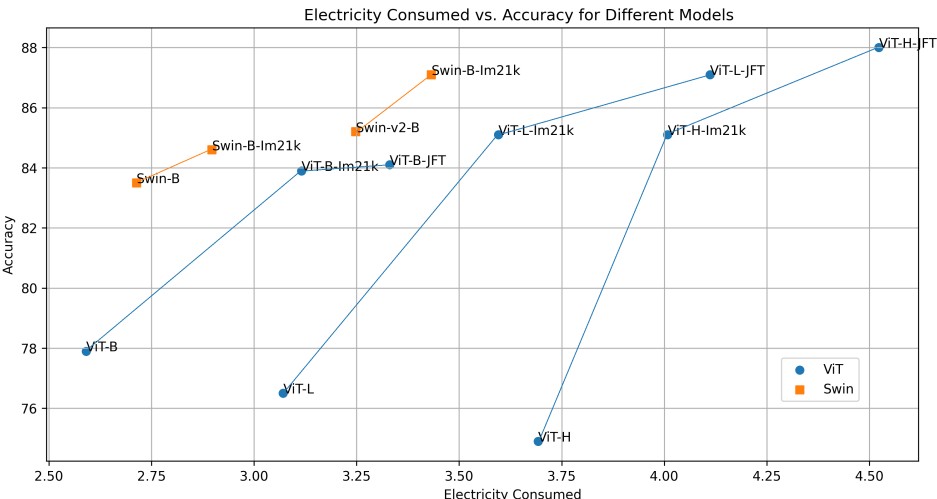

Figure 3: Comparing the effect of pre-training on Swin and ViT in terms of accuracy and electricity consumption.

pretraining on these datasets, and the environmental repercussions of such pre-training endeavors, we raise the fundamental question: is large-scale pre-training truly justifiable?

To address this query, we scrutinize the relative enhancements observed in models like Swin (trained on Im21k) and ViT (trained on JFT). In our exploration, as detailed in Appendix A.4.2, we identify a linear correlation between the volume of data and the projected electricity consumption for pre-training on these datasets. Our investigation delves into the performance shifts attributed to pre-training, as illustrated in Figure 3.

For instance, ViT-B exhibits a notable 6% performance enhancement compared to training from scratch, while requiring significantly fewer training epochs. However, it is important to note that achieving this 6% improvement consumes *ten times* the electricity as training from scratch. This cost-benefit analysis holds more weight for larger ViT models, which tend to exhibit underperformance when trained from scratch. While large-scale pre-training undeniably offers advantages across multiple tasks, our primary focus here pertains to image classification. Nonetheless, the crux of our inquiry is whether the electricity expended in this pursuit is commensurate with the gains achieved and if improved accuracies using publically unavailable datasets is fair for comparing against.

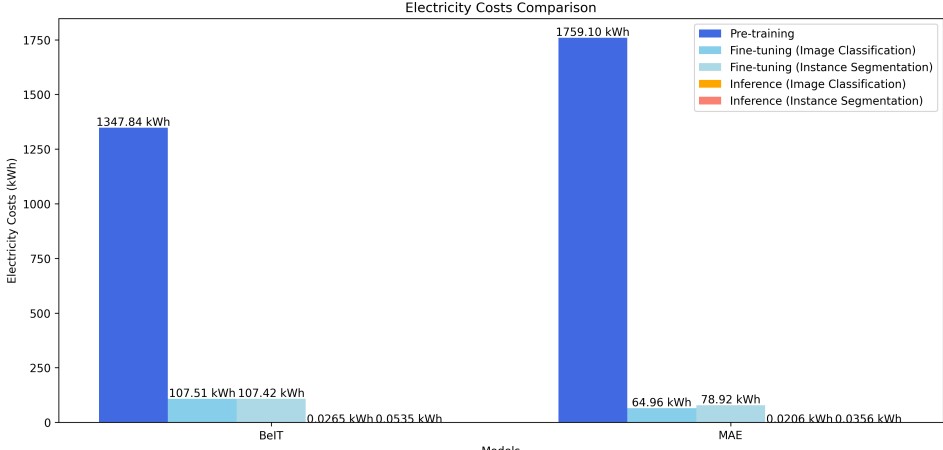

Figure 4: Comparing the cost of self-supervised pre-training and then fine-tuning along with inference of BEiT and MAE in terms of electricity consumption. The inference cost is so low in comparison that it is not even visible in the graph.

### 5.5 WHY IS SELF-SUPERVISED LEARNING USEFUL?

Self-supervised learning has been a major breakthrough in research because it empowers models to learn directly from vast amounts of unlabeled data, significantly reducing the dependency on expensive and time-consuming human annotation, and thereby accelerating progress in various domains. The representations learned through self-supervised learning can be leveraged across multiple downstream tasks, providing a versatile foundation that enables improved performance, transferability, and efficiency in various domains.

In our experiments, we explore two self-supervised methods, namely BEiT and MAE. Although self-supervised methods can be resource-intensive during training, as previously discussed in the context of large-scale pre-training, they offer the advantage of direct applicability to multiple downstream tasks. In our experiments, we specifically focus on image classification and semantic segmentation, however, MAE also reports on object detection.

In Figure 4, it's clear that self-supervised pre-training contributes to over 90% of the total training cost. This emphasizes the efficiency of fine-tuning these methods for various downstream tasks or datasets, highlighting the significant advantages of self-supervised learning in real-world applications. It suggests that self-supervised learning yields robust and transferable feature representations, making it highly valuable.

## 6 CONCLUSION

In conclusion, our study delves comprehensively into the critical intersection of deep learning model performance and energy consumption. While deep learning has unquestionably transformed numerous fields with its unparalleled accuracy, the escalating energy requirements of these models have given rise to environmental concerns and presented challenges for smaller research entities. To address this issue of escalating electricity consumption by models, we propose an innovative metric that places significant emphasis on accuracy per unit of electricity consumed. This metric not only levels the competitive field but also empowers smaller university labs and companies to compete effectively against their larger counterparts. Through an extensive examination of various deep learning models across different tasks, we have uncovered invaluable insights into the trade-offs between accuracy and efficiency. Our findings highlight the potential for more sustainable deep learning practices, where smaller, energy-efficient models can expedite research efforts while minimizing environmental impact. This research not only fosters a fairer research environment but also advocates for the adoption of efficient deep learning practices to reduce electricity consumption, ensuring a greener future for generations to come.

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

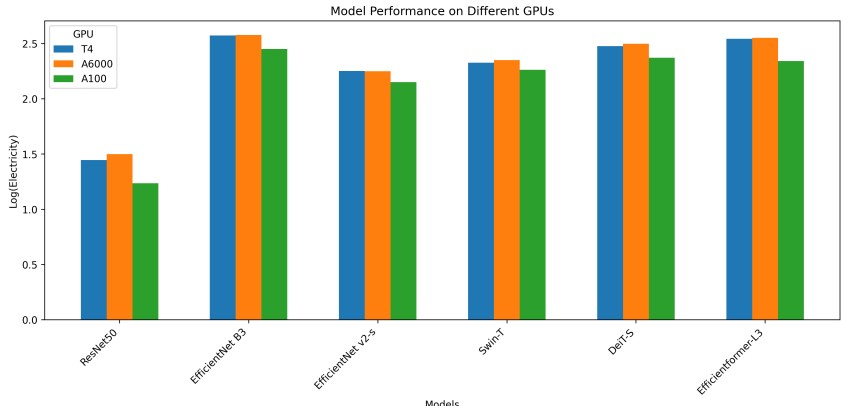

Figure 5: Model Performance Across GPUs: Despite Varied Hardware, Logarithmic Scaling Reveals Consistency

Hengshuang Zhao, Jianping Shi, Xiaojuan Qi, Xiaogang Wang, and Jiaya Jia. Pyramid scene parsing network. In *Proceedings of the IEEE conference on computer vision and pattern recognition*, pp. 2881–2890, 2017.

Sixiao Zheng, Jiachen Lu, Hengshuang Zhao, Xiatian Zhu, Zekun Luo, Yabiao Wang, Yanwei Fu, Jianfeng Feng, Tao Xiang, Philip HS Torr, et al. Rethinking semantic segmentation from a sequence-to-sequence perspective with transformers. In *Proceedings of the IEEE/CVF conference on computer vision and pattern recognition*, pp. 6881–6890, 2021.

Bolei Zhou, Hang Zhao, Xavier Puig, Sanja Fidler, Adela Barriuso, and Antonio Torralba. Scene parsing through ade20k dataset. In *Proceedings of the IEEE conference on computer vision and pattern recognition*, pp. 633–641, 2017.

Bolei Zhou, Alex Andonian, Aude Oliva, and Antonio Torralba. Temporal relational reasoning in videos. *European Conference on Computer Vision*, 2018.

## A   APPENDIX

### A.1   IMAGE CLASSIFICATION RESULTS

A detailed analysis of the projected numbers can be found in Table A.1. In the main paper, we show some of these results in Figure 2 for better visualization. However, we report the projected electricity consumption along with the GQI for various methods and their variants. As expected, methods that perform large-scale pre-training or self-supervised learning tend to use the most electricity.

### A.2   HOW MUCH DOES HARDWARE AFFECT ENERGY CONSUMPTION?

In evaluating the performance of various deep learning models on GPUs with different memory capacities, including the A100 with 40GB, Tesla T4 with 16GB, and A6000 with 48GB, an intriguing observation emerges. Despite the substantial variance in GPU memory, the utilization of a logarithmic scale, such as log(electricity) as a performance metric, effectively mitigates the discrepancies in raw performance metrics. This scaling allows for a fairer comparison, as it emphasizes relative improvements rather than absolute values. Consequently, even though the GPUs exhibit significant differences in memory size and hardware capabilities, the transformed data reveals that the models' performances remain remarkably consistent. This underscores the robustness of the chosen metric, which accommodates varying hardware configurations and ensures a reliable evaluation of deep learning model performance across diverse computing environments. This can be seen in Figure 6.

| Models | GFLOPs | Train Parameters | Electricity | Test Accuracy | GQI | Electricity |
|---|---|---|---|---|---|---|
| MobileNetv2 | 0.3 | 3.5M | 16.152 | 72.5 | 0.829 | 0.001622 |
| MobileNet v3 | 0.2 | 5.5M | 48.75 | 74 | 0.658 | 0.001427 |
| ResNet18 | 1.8 | 11.7M | 24.4134 | 69.7 | 0.593 | 0.005862 |
| ResNet50 | 4.1 | 25.6M | 31.506 | 76.1 | 0.852 | 0.009672 |
| ResNet101 | 7.8 | 44.5M | 41.694 | 77.4 | 0.858 | 0.012668 |
| ResNet152 | 11.5 | 60.2M | 55.6464 | 78.3 | 0.844 | 0.015516 |
| VGG16 | 15.5 | 138.4M | 50.0475 | 71.6 | 0.554 | 0.012264 |
| VGG19 | 19.6 | 143.7M | 55.9575 | 72.4 | 0.57 | 0.013352 |
| DenseNet121 | 2.8 | 8.0M | 55.91 | 74.4 | 0.653 | 0.010241 |
| DenseNet161 | 7.7 | 28.7M | 98.01 | 77.1 | 0.685 | 0.016241 |
| Inception v3 | 5.7 | 27.2M | 71.698 | 79.8 | 0.873 | 0.013816 |
| EfficientNet B0 | 0.4 | 5.3M | 222.894 | 77.1 | 0.581 | 0.011948 |
| Efficient Net B3 | 1.8 | 12.2M | 376.432 | 81.6 | 0.703 | 0.017964 |
| EfficientNet B7 | 37.8 | 66.3M | 1031.695 | 84.3 | 0.707 | 0.03876 |
| EfficientNetv2 S | 8.4 | 21.5M | 177.06 | 83.9 | 0.925 | 0.010084 |
| EfficientNetv2 M | 24.6 | 54.1M | 275.73 | 85.1 | 0.915 | 0.013908 |
| EfficientNetv2 L | 56.1 | 118.5M | 467.49 | 85.7 | 0.866 | 0.021164 |
| RegNetY-16GF | 15.9 | 84.0M | 136.05 | 80.4 | 0.788 | 0.018048 |
| ViT-B-scratch | 17.6 | 86.6M | 389.76 | 77.9 | 0.554 | 0.020572 |
| ViT-L-scratch | 61.5 | 304.3M | 1177.62 | 76.5 | 0.427 | 0.057576 |
| ViT-B-Im21k | 17.6 | 86.6M | 1304.656 | 83.9 | 0.668 | 0.020572 |
| ViT-L-Im21k | 61.5 | 304.3M | 3941.887 | 85.1 | 0.621 | 0.057576 |
| ViT-H-Im21k | 167.3 | 632.0M | 10188.512 | 85.1 | 0.557 | 0.155936 |
| ViT-B-JFT | 17.6 | 86.6M | 2139.002 | 84.1 | 0.632 | 0.020572 |
| ViT-L-JFT | 61.5 | 304.3M | 12925.5576 | 87.1 | 0.61 | 0.057576 |
| ViT-H-JFT | 167.3 | 632.0M | 33337.3983 | 88 | 0.584 | 0.155936 |
| Swin-T | 4.5 | 28.3M | 224.01 | 81.3 | 0.756 | 0.013468 |
| Swin-S | 8.7 | 49.6M | 357.318 | 83 | 0.772 | 0.019724 |
| Swin-B | 15.4 | 87.8M | 516.69 | 83.5 | 0.749 | 0.027082 |
| Swin-B+PT on Im21k | 15.4 | 87.8M | 1769.1457 | 85.2 | 0.692 | 0.027082 |
| Swin-v2-T | 5.9 | 28.4M | 307.044 | 82.8 | 0.783 | 0.020242 |
| Swin-v2-S | 11.5 | 49.7M | 493.512 | 84.1 | 0.782 | 0.029232 |
| Swin-v2-B | 20.3 | 87.9M | 789.69 | 84.6 | 0.748 | 0.042056 |
| Swin-v2-B+PT on Im21k | 20.3 | 87.9M | 2703.898 | 87.1 | 0.731 | 0.042056 |
| DeIT-T-300Ep | 1.3 | 6.0M | 244.194 | 74.5 | 0.481 | 0.004764 |
| DeIT-S-300Ep | 4.6 | 22.0M | 315.18 | 81.2 | 0.707 | 0.012444 |
| DeIT-B-300Ep | 17.6 | 87.0M | 527.436 | 83.4 | 0.742 | 0.020772 |
| DeIT-T-1000Ep | 1.3 | 6.0M | 496.53 | 76.6 | 0.49 | 0.004764 |
| DeIT-S-1000Ep | 4.6 | 22.0M | 733.15 | 82.6 | 0.671 | 0.012444 |
| DeIT-B-1000Ep | 17.6 | 87.0M | 1440.67 | 84.2 | 0.67 | 0.020772 |
| ViT-B-MAE | 17.6 | 86.6M | 1824.06 | 83.6 | 0.627 | 0.020572 |
| ViT-L-MAE | 61.6 | 304.3M | 2659.35 | 85.9 | 0.683 | 0.057576 |
| ViT-H-MAE | 167.3 | 632.0M | 6801.35 | 86.9 | 0.647 | 0.155936 |
| DaViT-T | 4.5G | 28.3M | 264.21 | 82.8 | 0.804 | 0.015764 |
| DaViT-S | 8.8G | 49.7M | 410.112 | 84.2 | 0.81 | 0.021372 |
| DaViT-B | 15.5G | 87.9M | 577.71 | 84.6 | 0.785 | 0.028792 |
| Clip-B/16 | 17.5G | 83.0M | 7697.28 | 86.6 | 0.627 | 0.029212 |
| Clip-L/14 | 80.7G | 506.0M | 20916.06 | 88 | 0.611 | 0.072422 |
| BEiT-L | 61.7G | 307.0M | 4025.685 | 85.2 | 0.623 | 0.072732 |
| BEiTv2-B | 17.6G | 86.0M | 2668.149 | 85.5 | 0.667 | 0.026452 |
| BEiTv2-L | 61.7G | 307.0M | 8135.588 | 87.3 | 0.649 | 0.071682 |
| EfficientFormer-L1 | 1.3G | 12.3M | 277.26 | 79.2 | 0.638 | 0.009224 |
| EfficientFormer-L3 | 3.9G | 31.3M | 356.01 | 82.4 | 0.745 | 0.013852 |
| EfficientFormer-L7 | 9.8G | 82.1M | 545.274 | 83.3 | 0.733 | 0.022664 |
| EfficientFormer-v2-s0 | 0.4G | 3.6M | 282.546 | 75.7 | 0.508 | 0.011532 |
| EfficientFormer-v2-s1 | 0.7G | 6.1M | 308.352 | 79 | 0.619 | 0.012544 |
| EfficientFormer-v2-s2 | 1.3G | 12.6M | 369.15 | 81.6 | 0.705 | 0.017844 |
| EfficientFormer-v2-l | 2.6G | 26.1M | 479.184 | 83.3 | 0.749 | 0.024532 |
| ConvNEXT-T | 4.5G | 29M | 308.508 | 82.1 | 0.75 | 0.007552 |
| ConvNeXt-S | 8.7G | 50M | 500.022 | 83.1 | 0.735 | 0.020024 |
| ConvNeXt-B | 15.4G | 89M | 714.384 | 83.8 | 0.725 | 0.027264 |
| ConvNeXt-L | 34.4G | 198M | 1239.204 | 84.3 | 0.689 | 0.045711 |
| ConvNEXT-T-Im22PT | 4.5G | 29M | 1056.331 | 82.9 | 0.648 | 0.007552 |
| ConvNeXt-S-Im22PT | 8.7G | 50M | 1712.0753 | 84.6 | 0.671 | 0.020024 |
| ConvNeXt-B-Im22PT | 15.4G | 89M | 2446.0508 | 85.8 | 0.687 | 0.027264 |
| ConvNeXt-L-Im22PT | 34.4G | 198M | 4243.0345 | 86.6 | 0.672 | 0.045711 |
| ConvNeXt-XL-Im22PT | 60.9G | 350M | 6300.9886 | 87 | 0.656 | 0.073708 |

Table 3: A comparison of electricity consumed, accuracy and the proposed metric over multiple models. All reported results are on ImageNet1K Deng et al. (2009).

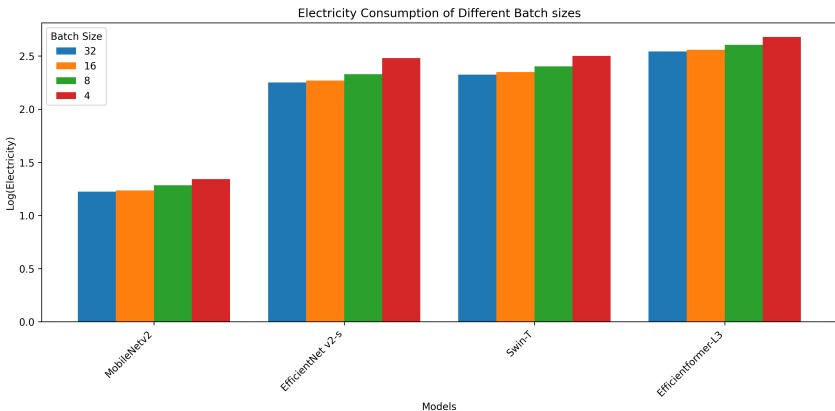

Figure 6: Model Performance Across Batch Sizes: The Need for Fixed Batch Sizes is Necessary for Fair Comparison.

### A.3 How do hyperparameters affect energy consumption?

To assess the electricity consumption of popular deep learning models, namely MobileNet v2, EfficientNet v2-s, Swin-T, and Efficientformer-L3, we conducted a comprehensive analysis by varying the batch sizes during testing. Our findings highlight an expected trend: as batch sizes decrease, electricity consumption increases. This phenomenon is rooted in the intricate dynamics of deep learning processes. Smaller batch sizes lead to reduced parallelization, resulting in longer training times and, consequently, higher electricity consumption. Notably, when examining these differences in electricity consumption on a logarithmic scale, the variations appear relatively modest. Nevertheless, it is imperative to emphasize that, for fair model comparisons using our metric, a fixed batch size must be maintained. This standardization is essential because our metric's basis is tied to electricity consumption, and any alteration in batch size could skew the results. In summary, while batch size alterations do influence electricity consumption, our metric's integrity hinges on maintaining a consistent batch size for equitable model evaluations. This is shown in Figure 6.

### A.4 How to scale up for quicker reporting of results?

One challenge associated with this metric is its susceptibility to hardware variations, where the efficiency may vary depending on the hardware used (for instance, TPUs might offer better performance, or larger GPUs might allow for larger batch sizes). Consequently, if we were to assess our models against this metric, we would need to re-run them multiple times using different hardware configurations. However, our research demonstrates that this dependency is actually linear with respect to the number of epochs and the percentage of data utilized. This means that we can initially run each model for just one or five epochs on a small subset of data, such as 1% or 10%, and then easily extrapolate the results to the entire dataset and the specified number of epochs mentioned in each research paper.

#### A.4.1 Dependency of energy consumption with number of epochs

We run MobileNet v2, ResNet50, EfficientNet B3, EfficientNetv2 M, ViT-B and EfficientFormer-L3 for a total of 20 epochs and calculate electricity consumption at each epoch. We plot this in Figure 7. We see an approximately linear relationship and hence can easily scale up to any number of epochs. Further, since we use log scale in the metric, any small variations are further diminished. We use 1% of the overall training data of ImageNet1k.

#### A.4.2 Dependency of energy consumption with percentage of data

We run MobileNet v2, ResNet50, EfficientNet B3, EfficientNetv2 M, ViT-B and EfficientFormer-L3 for a total of 20 epochs and calculate electricity consumption at each epoch, but in this case we run them on varying amounts of data starting from 1% then 10%, 20%, 50% and finally 100% of the

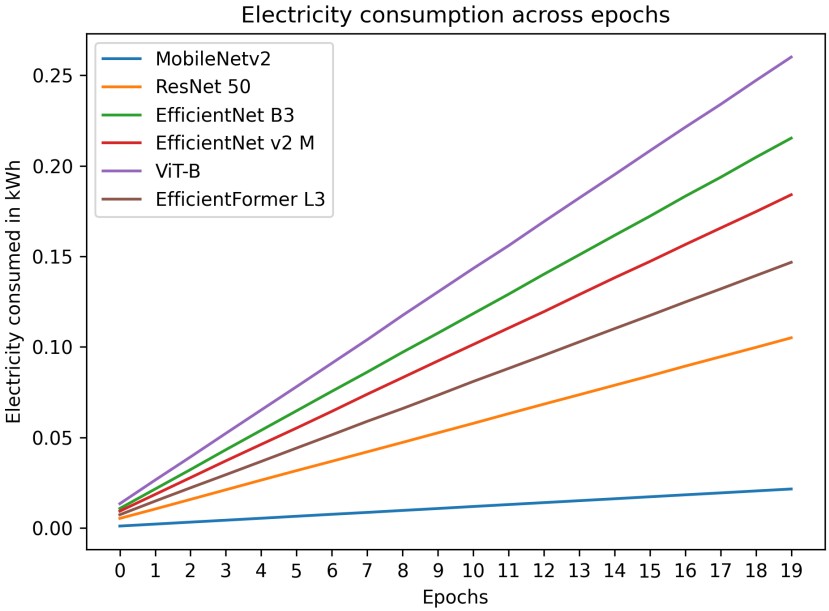

Figure 7: Electricity Consumed Across Epochs: Linear Relationship Observed.

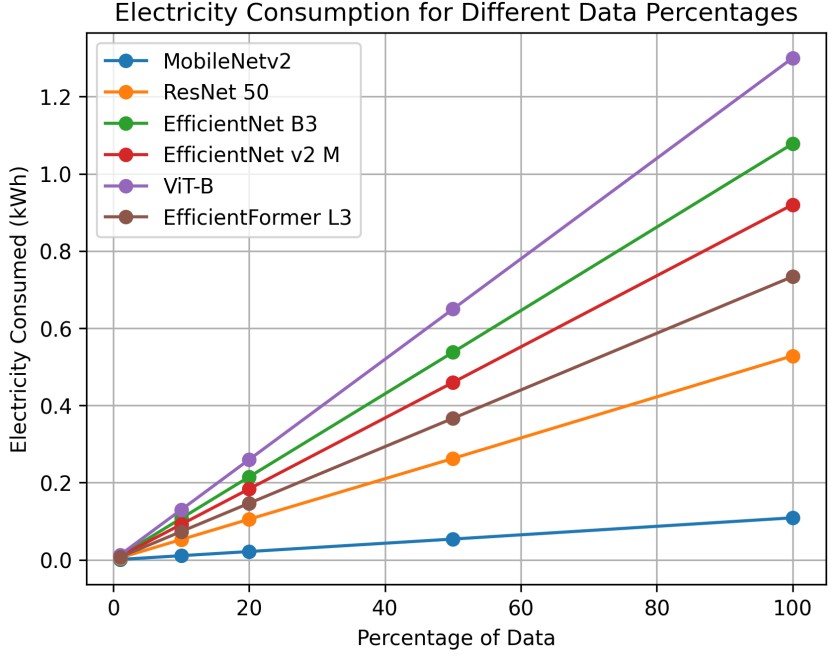

Figure 8: Electricity Consumed Across Different Percentages of Data: Linear Relationship Observed.

data. We plot this in Figure 8. We see an approximately linear relationship and hence can easily scale up to 100% of the data with just 1% of the data. Further, since we use log scale in the metric, any small variations are further diminished.

### A.4.3 WHY APPROXIMATIONS WORK?

The practice of approximating electricity consumption values by running models on a fraction of the dataset and for a fraction of the total epochs is efficient and viable due to several key factors. Firstly, the observed approximately linear relationships between energy consumption and both the number of epochs and the percentage of data used allow for straightforward extrapolation, enabling reliable estimates. Additionally, the use of a logarithmic scale in the energy consumption metric mitigates the impact of minor variations, ensuring the robustness of these approximations. Moreover, this approach significantly reduces computational load and time requirements, making it resource-efficient, especially when dealing with extensive datasets or rapid model configuration assessments. Lastly, the scalability of these approximations facilitates their application in scenarios involving larger datasets or extended training periods, further enhancing their practical utility.

## A.5 HYPERPARAMETERS OF THE METRIC

In the context of image classification, we established fixed values of $\alpha$ and $\beta$ as 5. What we observed was that, before setting $\alpha$ to 5, MobileNet v2 consistently achieved the highest GQI value, regardless of more recent research findings. This was primarily attributed to MobileNet v2's notably low power consumption. However, upon setting $\alpha$ to 5, we found that EfficientNet v2 surpassed all others in terms of GQI, while methods such as Swin and DeiT were competitive at a similar level.

When it came to setting $\beta$, our focus shifted to examining the GQI values when $\beta$ was set to 1. Within this range, GQI values fluctuated between 0.111 and 0.185, presenting challenges in terms of interpretability. Consequently, we decided to increase the value of $\beta$ to 5. This adjustment led to a narrower GQI range, now spanning from 0.555 to 0.925, significantly enhancing the interpretability of the results.

We ultimately opted to maintain these $\alpha$ and $\beta$ values consistently for both action recognition and semantic segmentation tasks, and our reported results in the tables reflect this choice.

