# OpenReview forum: "WATT FOR WHAT: RETHINKING DEEP LEARNING’S ENERGY-PERFORMANCE RELATIONSHIP"
_ICLR.cc/2024/Conference — ICLR 2024 Conference Withdrawn Submission_

### Official Review · Reviewer_MAKR · 2023-10-31

**Soundness:** 1 poor
**Presentation:** 2 fair
**Contribution:** 2 fair
**Rating:** 3
**Confidence:** 5

**Summary:**

The Paper “WATT FOR WHAT: RETHINKING DEEP LEARNING’S ENERGY-PERFORMANCE RELATIONSHIP” addresses the problem of increasing energy consumption and subsequent Carbon footprint of today’s deep learning research.
The authors introduce a novel metric, the GreenQuotientIndex (GQI), which aims to compare machine learning models based on their accuracy and energy efficiency. According to the authors, GQI penalizes models for their excessive electricity usage, with the penalty increasing logarithmically, while also rewarding models for achieving higher accuracy.
The authors then present a number of experiments to compare different deep learning architectures on computer vision tasks by using the proposed metric. They partially train various CNN and Transformer-based existing models on image classification, semantic segmentation and action recognition and track the electricity consumption with CodeCarbon, an open-source python tool.

While the topic is very important the novelty of the paper is very limited. Given that the authors did not conduct full training runs with extensive analysis of the energy consumption and corresponding predictive performance, the contribution of the paper boils down to a metric, that has several flaws (see Weaknesses).

In general I am not a fan of merging energy consumption and model predictive performance into one metric, as it smears things out. If Model A has a GQI 0.1 higher than Model B, it's unclear what this difference means in terms of their performance or efficiency. Furthermore, two models can have the same GQI, but with different values of Accuracy and Energy consumption. Depending on the application, either one would be the better choice (High model performance e.g. in security settings for face recognition vs low energy consumption for embedded devices. It would be much better if both values were to be reported individually, such that researchers can evaluate the full extend and implications of the model themselves. As claimed by current research, it is more important to attribute the same scientific value to improvements in energy efficiency as in predictive performance.

**Strengths:**

- The topic on energy consumption and related carbon footprint of deep learning models is very timely and needs to be addressed urgently
- For Image recognition, the experimental evaluation in terms of different models is quite extensive, including different architectural components.
- The introduction is quite nicely written, though it is more adequate for a comment than a research paper and could do with a few more proofs to its claims.

**Weaknesses:**

The proposed metric has several flaws:
- The hyperparameter ß seems useless, as its sole purpose appears to be scaling to obtain “nice” values (the authors claim the purpose of “better understanding of the values”, but given that the metric has no physical meaning and hyperparameters are chose quite arbitrary, there is no point to that, and absolute values of the metric have no meaning whatsoever. Energy scale is comparable if it is fixed for all investigated cases)
- Using the log of the energy consumption and the power of the accuracy has exactly opposite effects on the resulting metric as intended. The authors chose accuracy^a to reward high accuracies. But their own experiments demonstrate that achieving the last few percent in accuracy gain is the most costly in terms of electricity consumption. And on the other hand, scaling the electricity consumption with the log penalizes smaller, more energy efficient models much harder, than those with excessive energy consumption. For example, the GQI of ViT-H-JFT and ViT-B-scratch is roughly the same, but the energy consumption of the Former is two orders of magnitude larger, while the accuracy difference is only ~10%.
- The hyperparameters an and B are tuned solely to promote certain models (EfficientNet), because for other values, MoblieNet always achieved the highest GQI. This is highly non-scientific.
- The properties of the metric: in Sec 3.1 in the first bullet point, the authors claim the nominator to be in the range [0,inf], and directly following up in the next bullet point state that it is the power function of a number smaller then 1, directly contradicting the first statement. Moreover, the denominator cannot be 0, thereby rendering the statement on its range false as well. Furthermore the claim on the importance of the metric being non-negativeis nonsense. A metric can also be e.g. between -1 and 1 and would make sense. Since the metric has no physical meaning or units, the scale can be completely arbitrary, just like MSE for example.


The experimental evaluation:
- Only computer vision tasks based on images are investigated. In the introduction, the authors clearly discuss that current LLMs are the prime energy consumers. Hence, it would be good to include NLP tasks in the evaluation, especially with respect to the corresponding metrics, such as perplexity, BLUE, or ROGUE.Also, models whose predictive performance is not evaluated on an accuracy between 0 and 1, e.g. regression tasks with MSE/MAE
- There is no proper description of the experiments (see questions)
- While presenting the results, no attempt is made to gain insight into why specific architectures are more efficient than others.
- All experiments disregard data-parallel training (which cannot be assessed using CodeCarbon), but this is an essential part of DL training these days.

Certain assumptions (in the appendix):
- Interpolation: This is dangerous, since the number of epochs for each model can vary gravely. Even if you assume the number of epochs stated in the original papers (which are not always clearly given), this does not factor in things like early stopping. Moreover, the considered number of samples, batch size and other hyperparameters are strongly linked and can gravely affect model convergence.
- The section on hardware effects is pointless. The authors state that the substantial variance in GPU memory are mitigated by using the log of the electricity consumption. But it is well known that GPUs of newer generations (the ones with bigger memory) are also designed under energy efficiency aspects, which is likely to play a much more important role.

Presentation: Though nicely written, the graphical presentation of the paper is really poor:
- The Figures are terrible to read in print-out
- The table references and captions do not match, e.g. Table 1 is referred to as Table 5.2, Table 2 is referred to as Table 5.3, etc.
- Table layouts lack uniformity, even when their content is similar.
- In the description of Fig.2  x- and y-axis are swapped.
- The separation in Training and Test in Tables 1 and 3 is unclear
- All tables and plots match units for electricity
- Figure 1 is nonsensical, it seems like a rather half-baked comparison

**Questions:**

Please describe the experiments more clearly:
- You do not mention the number of epochs you train models on image recognition. In the Appendix you state interpolation from just 1 to 5 epochs?
- If you did not fully train those models, how do you obtain the final accuracy?
- Most experiments, I assume, entail fine-tuning pre-trained models on the mentioned datasets? Though it is very poorly explained.
- Table 1: should 'Train' and 'Test' be over the columns 'Electricity'?
- How do you measure GFLOPs in Table 1 and 3
- It is unclear whether reported GQI is based on training or test electricity consumption

---

### Official Review · Reviewer_JEc8 · 2023-10-31

**Soundness:** 2 fair
**Presentation:** 2 fair
**Contribution:** 1 poor
**Rating:** 3
**Confidence:** 4

**Summary:**

The paper proposed an expression combining accuracy and electrical energy use into a single scalar for use in single-objective comparison of ML models and evaluate several existing models using this expression.

**Strengths:**

+ Jointly considering the accuracy and efficiency implications of model design decisions is important, and is rarely done well in AI/ML research. The authors have selected an important problem to work on.

+ The article provides a decent survey / tutorial of research on model efficiency and environmental impact.

+ Stating the desired properties of the metric formally, in a list, is a very good approach (Section 3.1).

+ It would be nice to have a generally agreed upon metric considering energy use and accuracy so it wouldn't be necessary to hold one of the two roughly constant when experimentally comparing models.

+ The comparisons given by the author would be quite useful to those for whom the proposed metric matches their own.

**Weaknesses:**

+ There isn't anything fundamental about the expression the authors have proposed. It has some nice properties, but so do other members of a family of functions. It isn't even clear that a scalar is needed for comparisons. Multiobjective comparisons are possible, too, and the appropriate trade-offs depend on the application. The authors have essentially taken on the problem of considering the generally accepted value of accuracy and energy on the same scale, which is an important problem, but it is not clear that their expression solves this problem.

+ I think there might be some dropped units in the GQI expression. If GQI is to be unitless, Beta needs to compensate for log10(kW-h).

+ The text in Figure 2 is so small that it is impractical to read. It looks like a 3pt or 4pt font.

+ Not sure where to put this. It's not really a weakness, more a comment intended to indicate what would make this work great. If the authors could find a broadly accepted relationship between the value of energy and the value of accuracy that holds for many applications, that would be important, high-impact work.

**Questions:**

1) How much difference is there between electrical energy and latency? In other words, will work considering the trade-off between latency and accuracy lead in the same direction as yours, which considers the trade-off between energy and accuracy?

2) Why not contrast models using a multi-objective cost function instead of reducing to a scalar? Concepts like Pareto-ranking enable this.

3) Why does the fact that going from 80%-90% accuracy is generally more difficult than going from 30%-40% accuracy imply that the metric should raise accuracy to an exponent? Is 10% inherently more valuable in most applications near the top of the range than near the bottom of the range? Why isn't 10% improvement equally valuable everywhere in the range? More precisely, why are you implicitly taking into account model cost-to-implement using an arbitrary exponent in devising a metric that should reflect only model value, instead of just explicitly considering the cost-to-implement through the energy component?

4) Same question for justification of taking the log of energy use. Isn't a kW-h a kW-h in terms of cost and environmental impact?

---

### Official Review · Reviewer_nMed · 2023-11-01

**Soundness:** 1 poor
**Presentation:** 3 good
**Contribution:** 1 poor
**Rating:** 1
**Confidence:** 4

**Summary:**

The authors take a look at the tradeoff between energy costs and model accuracy in a variety of neural networks. They propose a unified metric to summarize this trade-off (the "GreenQuotientIndex"), report estimated electricity costs using an off-the-shelf software energy estimation library, and make some observations on the results.

**Strengths:**

- The authors are right to consider energy tradeoffs. Accuracy does not exist in a vacuum, and quantitative numbers on what accuracy costs is a laudable goal.

**Weaknesses:**

- It's unclear what the paper is actually contributing. Their GQI metric seems arbitrary and lacks a clear use case, their experimental methodology is inconsistent and narrow, and their conclusions seem to be extremely broad questions rather than quantifiable statements. E.g.: "is large-scale pre-training truly justifiable?" Yes, obviously, to those footing the bill. Are the externalities in carbon footprint damaging to others? Also yes, and that raises some tough questions. This issue is not new, but the paper doesn't seem to add anything novel to the discussion.
- The experimental methodology is weak. The authors' use of downsampling and epoch limiting is not fundamentally problematic, but they ignore a lot of nuance in scale. For instance, distributed training comes with costs, data centers have different power distribution and cooling efficiencies, etc.
- The authors ignore non-renewable electricity costs (capex). This has been shown to be non-negligible and, in some cases, dominates operational costs. (E.g.: Gupta, et al. "Chasing Carbon: The Elusive Environmental Footprint of Computing", 2021)
- The authors vastly oversimplify the fundamental tradeoffs in computational cost and accuracy. They completely ignore the economics of these decisions (if someone is willing to pay for dozens of MWh to train a model, they're not doing it blindly---energy is not cheap).
- The conclusions in Appendix 2 are *starkly* incorrect. There is a huge literature on energy-efficient accelerators for neural nets which refutes this directly, not to mention the entire commercial sector built on top of the idea that specialized hardware *does* substantially affect energy consumption.

**Questions:**

N/A

---

### Official Review · Reviewer_MKQf · 2023-11-06

**Soundness:** 2 fair
**Presentation:** 3 good
**Contribution:** 2 fair
**Rating:** 5
**Confidence:** 3

**Summary:**

This paper introduces a new metric GQI to show the trade-off between performance and environmental sustainability. Multiple experiments on different tasks, pre-training vs. training from scratch, and self-supervised learning show the importance of such a metric.

**Strengths:**

1. The paper provides a thorough related work section to motivate the importance of including electricity into consideration.
2. The experiments demonstrate the necessity of introducing the new metric.

**Weaknesses:**

The contribution is limited.

1. The GQI only considers electricity. As the paper cited, the life cycle assessment (LCA) is a comprehensive evaluation rather than electricity to measure the environmental impact, since LCA includes hardware manufacturing, transportation, and recycling. In addition, a recent paper [1] shows that the manufacturing carbon may be even higher than electricity carbon.
2. GQI's electricity estimation is from the existing tool CodeCarbon, which also limited this paper's contribution. In addition, there existing metrics that consider the more comprehensive electricity consumption. For example, PUE [2] is a ratio that measures how much energy is directly used for computing rather than for cooling and other support systems in a data center.

[1] Gupta, Udit, et al. "ACT: Designing sustainable computer systems with an architectural carbon modeling tool." Proceedings of the 49th Annual International Symposium on Computer Architecture. 2022.
[2] Avelar, Victor, et al. "PUE: a comprehensive examination of the metric." White paper 49 (2012).

**Questions:**

1. Can the author make the GQI more comprehensive? For example including the LCA and PUE.